# Factors Affecting Willingness to Receive Online Counseling: The Mediating Role of Ethical Concerns

**DOI:** 10.3390/ijerph192416462

**Published:** 2022-12-08

**Authors:** Xin Chen, Anzheng Du, Rufang Qi

**Affiliations:** School of Psychology, Henan University, Kaifeng 475000, China

**Keywords:** self-stigma, online interpersonal trust, ethical concerns, online counseling willingness

## Abstract

Since the outbreak of the COVID-19 pandemic, traditional face-to-face counseling has gradually given way to online counseling. To improve the application value of online counseling and change the current situation of college students’ lack of willingness to receive online counseling, this study explored factors that influence Chinese college students’ willingness to receive online counseling (WROC). Based on data gathered from surveying 823 Chinese college students using self-report questionnaires, we clarified the relationships between the self-stigma of seeking help, ethical concerns about online counseling (ECOC), online interpersonal trust (OIT), and the willingness to receive online counseling (WROC). The results indicated that (1) self-stigma of seeking help and OIT negatively and positively predicted the WROC, respectively; (2) ethical concerns negatively predicted the WROC; and (3) ethical concerns mediated the relationship between self-stigma and WROC and between OIT and WROC. The results suggest that reducing the self-stigma surrounding help-seeking, perfecting the ethical norms of online counseling, and enhancing interpersonal trust can improve willingness to receive online counseling.

## 1. Introduction

In recent years, with the rapid growth of interpersonal voice and video communication software, online psychological counseling has come into the spotlight; it is also known as online, remote, network, or electronic counseling. It involves qualified psychological counseling staff who establish a trusting relationship with help-seekers through online communication tools such as email, online chat rooms, and instant voice or video; they provide professional services, including psychological counseling and therapy [1,2]. Compared to face-to-face counseling, it is quick, accessible, flexible, economical, and not limited by a specific time or place [3]. Online counseling is as effective as face-to-face counseling in the treatment of psychological disorders such as depression, anxiety, and bipolar disorder [4,5,6].

Since the beginning of the COVID-19 pandemic, there has been a certain risk of viral contagion through physical contact, which has led to a shift from traditional face-to-face counseling to the currently predominant online format. Many mental health mobile applications that can help users relieve psychological stress, maintain mental health, and compensate for face-to-face counseling when necessary have also emerged. A recent study by Yuduang et al. [7] found that people have little knowledge of the mental health services available online and that increasing awareness of online mental health services and understanding their benefits and accessibility would increase their willingness to use them. Therefore, studying the factors that influence people’s willingness to use online mental health services can help to better utilize online service platforms to promote mental health.

The younger generation has grown up with the internet and is familiar with the online environment and interpersonal communication software. However, surveys and qualitative interviews have found that college students’ willingness to receive online counseling is not high [2,8]. This reluctance is mainly owing to some of their specific concerns about online counseling, such as: online counselors violating counseling ethics and divulging personal counseling information; personal privacy being violated by network technology, resulting in stigmatization; and the perceived difficulty in obtaining timely help from online counselors in the event of a psychological crisis [8,9,10]. Therefore, it is in line with practical needs and social development trends to study the factors that affect college students’ willingness to seek online counseling. This would aid in improving the application value of online counseling and change the current situation of college students’ lack of willingness to receive it.

Ajzen’s [11] theory of planned behavior underlines factors that may affect behavior indirectly through behavioral intention. Three cognitive factors—attitudes, subjective norms, and perceived behavioral control—play an important role in explaining behavioral intention and subsequent actions. People are more likely to engage in a behavior if they perceive it as useful to do so (attitudes), if their views are affirmed by those who value them (subjective norms), and if they have the necessary resources and opportunities to engage in that behavior (behavioral control). Based on this theory, the aim of the present study was to deepen the understanding of people’s conduct in seeking online counseling by examining their willingness to receive it.

The willingness to receive online counseling (WROC) is related to people’s evaluations, internal feelings, and acceptance of online counseling, as well as whether they choose this method to deal with psychological problems. Prior to the COVID-19 pandemic, Huang et al. [12] found that college students’ attitudes toward online counseling were not positive. Therefore, what are people’s attitudes toward online counseling in the post-pandemic era? Which factors influence a person’s WROC?

### 1.1. The Self-Stigma of Seeking Help and WROC

The existing literature divides help-seeking stigma into two categories: public stigma and self-stigma. The self-stigma of seeking help emerges when individuals who need to seek psychological services label themselves as socially unacceptable due to experiencing a psychological disorder or distress; they believe that seeking counseling services will lead to discrimination by others, thereby lowering their self-esteem and self-worth [13]. Such negative beliefs tend to reduce the willingness to seek counseling [14]. Research has demonstrated a significant, negative relationship between the self-stigma of seeking help and the willingness to receive face-to-face counseling [15,16], but how does the self-stigma of seeking help impact WROC? Joyce [17] suggested that online counseling provides greater privacy and may alleviate the sense of embarrassment one might experience when receiving face-to-face counseling. Further, individuals are more likely to have a positive attitude toward online counseling. Whether the self-stigma of seeking help influences the WROC in a positive or negative way, there have been few studies on this issue in the Chinese context. As such, it is necessary to provide fresh evidence for the relationship between the two phenomena. Hence, we postulated the following:

**Hypothesis 1.** 
*The self-stigma of seeking help can directly predict WROC.*


### 1.2. Online Interpersonal Trust and WROC

Online interpersonal trust (OIT) is a generalized expectation of an individual’s reliability based on the words, promises, and written or oral statements of the object of communication during risky online interpersonal interactions [18]. Internet communication has the advantages of speed, immediacy, and breakthrough in time and space, but its virtuality, fragility, and uncertainty reduce its trustworthiness and authenticity. A person’s OIT level also varies due to different personalities, online usage environments, and other factors; a low OIT reduces individuals’ willingness to socialize online [19]. Online psychological counseling needs to be realized through social tools such as email, voice, or video networks, because it is both a form of psychological counseling and a special online social activity. As such, we formed our second hypothesis:

**Hypothesis 2.** 
*The degree of OIT can directly predict WROC.*


### 1.3. Ethical Concerns about Online Counseling: A Mediating Role

Online counseling ethics provide the moral code of conduct that counselors and clients follow in remote psychological counseling. They embody a system of value judgments and behavioral concepts that integrate virtual and real situations and connect both parties. Online counseling ethics play an important role in standardizing counselors’ behavior and protecting clients’ interests [20]. Ethical concerns about online counseling (ECOC) involve clients’ skepticism about counselors’ compliance with ethical norms and the correct handling of ethical dilemmas when seeking online counseling, such as concerns about the confidentiality of network technology and the interview environment.

People who turn to online counseling may be worried about its safety, and their satisfaction with and expectations of online counseling are lower than those when receiving face-to-face counseling [21]. At present, most research on online counseling ethics in China focuses on the counselor level and pays less attention to help-seekers’ ECOC. In addition, there are questions about whether people with a high degree of self-stigma will have a stronger ECOC in order to protect their self-esteem and sense of self-worth and whether people with a high degree of interpersonal trust on the internet will have fewer ethical concerns because they are more aware of online consultations. Thus, the degree of ECOC may affect WROC, but relevant studies have rarely been published. Hence, we proposed the following hypothesis.

**Hypothesis 3.** 
*ECOC plays a mediating role in the relationship between self-stigma and WROC and between OIT and WROC.*


## 2. Materials and Methods

### 2.1. Participants and Procedures

#### 2.1.1. Participants

The participants were 823 college students recruited through convenience sampling. Most of the participants were from Henan Province, with a few from other provinces. They were between 18 and 24 years old. The sample comprised 386 men (46.90%) and 437 women (53.10%); 466 (56.6%) were majoring in science and technology, and 357 (43.4%) studied other majors.

This study was approved by the Institutional Review Board of Henan Province Key Laboratory of Psychology and Behavior (protocol code: 20210911001). Participants gave consent for their data to be used in the research.

#### 2.1.2. Procedures

We collected the data by handing out paper questionnaires in November 2021 and administering an online survey in December 2021. Trained research assistants introduced the study to the participants and obtained their verbal informed consent. Participants were told they could withdraw at any time and were instructed to complete the survey independently. The paper questionnaires were handed out in the classroom, and we obtained informed consent from the institution in which this classroom was located. The online survey was administered through the website Questionnaire Star. After the questionnaires were collected, we excluded the incomplete ones and those with obviously false answers, finally ending up with 90.2% valid questionnaires.

### 2.2. Measurements

#### 2.2.1. WROC

We used the Online Counseling Attitudes (OCA) scale, revised by Rochlen [22], to assess WROC. The OCA is a 10-item self-report instrument that measures evaluations of online counseling (e.g., “Using online counseling will help me learn about myself”) and discomfort with online counseling (e.g., “I dread explaining my problems to an online counselor”). Participants respond to the items on a 6-point Likert scale ranging from 1 (strongly disagree) to 6 (strongly agree); higher scores denote more positive evaluations of, or a stronger discomfort with, online counseling. After we reverse-scored the items of discomfort with online counseling, higher total OCA scores indicated a greater WROC. The Cronbach’s α coefficients for the evaluations of online counseling and discomfort with online counseling were 0.854 and 0.813, respectively.

#### 2.2.2. Self-Stigma of Seeking Help

The Self-Stigma of Seeking Help (SSOSH) scale was created by Vogel et al. [23] and is a 10-item, self-report instrument measuring self-stigma (e.g., “My self-esteem would increase if I talked to a therapist”). Participants respond to the items on a 5-point Likert scale ranging from 1 (not at all) to 5 (extremely well). Higher scores indicate a stronger self-stigma of seeking help. In this study, the Cronbach’s α coefficient for the SSOSH scale was 0.73.

#### 2.2.3. ECOC

We created the Ethical Concerns About Online Counseling scale (ECOCS) according to Section 8 of the Code of Work Ethics about Clinical and Consulting Psychology, 2nd edition [24]. The ECOCS is a 14-item, self-reporting tool designed to gauge the level of ECOC along three dimensions, including concerns about informed consent and confidentiality, the counseling relationship, and crisis intervention. Participants respond to the items on a 6-point Likert scale ranging from 1 (strongly disagree) to 6 (strongly agree); higher scores indicate a greater ECOC. The Cronbach’s α coefficient for the ECOCS was 0.879. The Cronbach’s α coefficients for the three dimensions of this scale were 0.901, 0.710, and 0.697 for informed consent and confidentiality, the counseling relationship, and crisis intervention, respectively.

#### 2.2.4. OIT

The Online Interpersonal Trust (OIT) scale was created by Ding and Shen [25] and was used to determine the extent of the participants’ online interpersonal trust. The OIT scale has 10 items (e.g., “I believe that most of the information provided by online friends is true”), and the original scale uses a 5-point Likert scale to respond to the items. However, to avoid the central tendency, we used a 6-point Likert scale ranging from 1 (strongly disagree) to 6 (strongly agree); higher scores denote higher levels of OIT. The Cronbach’s α coefficient for the OIT scale in the current study was 0.706.

### 2.3. Analysis

We used descriptive statistics and correlation and regression analysis to calculate the means and standard deviations of the main variables. We analyzed the relationships between the variables using IBM SPSS Statistics (Version 21.0, IBM, Armonk, NY, USA). IBM SPSS Amos (Version 26.0, IBM, Armonk, NY, USA) was used to build the mediation model and to provide the fit indices, the coefficient and significance testing, and the bootstrap confidence interval (95%, sample: 2000).

## 3. Results

### 3.1. Descriptive Statistics and Correlations among the Variables

Table 1 presents the means, standard deviations, skewness, and kurtosis of all variables and the results of the correlation analysis. The skewness and kurtosis of all variables were between negative 1 and positive 1, which indicates that each variable obeys the normal distribution. The correlation coefficients indicated that online counseling attitudes were negatively related to the self-stigma of seeking help (*r* = −0.30, *p* < 0.01), positively related to OIT (*r* = 0.24, *p* < 0.01), and negatively related to ECOC (*r* = −0.18, *p* < 0.01). That is, the stronger the self-stigma of seeking help, the lesser the WROC, and the higher the OIT, the greater the WROC. Meanwhile, a greater ECOC was related to a lesser WROC. Additionally, ECOC was positively related to the self-stigma of seeking help (*r* = −0.20, *p* < 0.01) and negatively related to OIT (*r* = −0.10, *p* < 0.01). OIT was not significantly linked to the self-stigma of seeking help (*r* = −0.03, *p* > 0.05). This result implies that a higher level of self-stigma of seeking help leads to a greater ECOC. However, the higher the OIT, the lesser the ECOC. There was no significant relationship between OIT and self-stigma of seeking help (*r* = −0.03, *p* > 0.05).

### 3.2. The Direct Effects of the Self-Stigma of Seeking Help and OIT on Online Counseling Attitudes

We used linear regression to evaluate the predictive effect of the self-stigma of seeking help and OIT on online counseling attitudes. Table 2 presents the results of the linear regression analysis. The regression model was significant (*R*^2^ = 0.14, *F* = 66.21, *p* < 0.01), and the self-stigma of seeking help significantly and negatively predicted online counseling attitudes (β = −0.29, *p* < 0.01). That is, a high level of self-stigma of seeking help can reduce the WROC. However, OIT significantly and positively predicted online counseling attitudes (β = 0.23, *p* < 0.01). Hence, a high level of OIT can increase the WROC. 

### 3.3. The Mediation Effects of the Three Dimensions of ECOC

We used structural equation modeling to assess the model of the mediation effects of the three dimensions of ECOC. When forming the specifications of the model, we assumed there would be a correlation among the three dimensions of the ECOC residual errors. The model fit indices of the research models demonstrated adequate goodness-of-fit statistics (χ^2^*/df* = 2.841, *p* = 0.092, RMR = 0.009, GFI = 0.999, AGFI = 0.972, NFI = 0.997, RFI = 0.939, TLI = 0.960, IFI = 0.998, CFI = 0.998, RMSEA = 0.047). 

As shown in Figure 1, the self-stigma of seeking help did not directly predict OCA1 (β = −0.05, *p* > 0.05) but directly and positively predicted OCA2 (β = −0.33, *p* < 0.01). This implies that a strong self-stigma of seeking help can increase discomfort with online counseling. OIT directly and positively predicted OCA1 (β = 0.21, *p* < 0.01) and negatively predicted OCA2 (β = −0.10, *p* < 0.05). This suggests that higher levels of OIT can increase the positive evaluation of online counseling and reduce discomfort with online counseling.

The indirect effects of ECOC1, ECOC2, and ECOC3 were indicated by a bias-corrected bootstrap at the 95% confidence interval (CI). Hence, self-stigma of seeking help was significantly and indirectly related to OCA1 through ECOC1, ECOC2, and ECOC3. The standardized indirect effect was 0.0396 (95% bootstrap CI: 0.010, 0.072). However, OIT was not indirectly related to OCA1 through ECOC1, ECOC2, or ECOC3. The standardized indirect effect was 0.0045 (95% bootstrap CI: −0.019, 0.025). Meanwhile, self-stigma of seeking help was significantly and indirectly related to OCA2 through ECOC1, ECOC2, and ECOC3. The standardized indirect effect was 0.0614 (95% bootstrap CI: 0.032, 0.091). OIT was significantly and indirectly related to OCA2 through ECOC1, ECOC2, and ECOC3. The standardized indirect effect was −0.0206 (95% bootstrap CI: −0.041, −0.001).

## 4. Discussion

### 4.1. Self-Stigma of Seeking Help Reduced WROC

We found a significant negative correlation between self-stigma of seeking help and WROC; that is, the stronger the individual’s perceived self-stigma, the lesser the WROC, which supports Hypothesis 1. This result is consistent with the finding that the self-stigma of seeking help reduces the willingness to receive face-to-face counseling [14,16]. Individuals with a high level of self-stigma are fearful that they will experience lower self-worth by seeking professional psychological counseling, and they will thus avoid or be ashamed of seeking help. 

The general public is biased against seeking help for psychological problems and characterizes people who do so as having personality defects, poor communication skills, and strange behavioral tendencies [15]. When help-seekers are affected by social prejudice and negative representations, they become less motivated to seek help and avoid professional psychological assistance to prevent negative evaluations, negative self-image, and marginalization through the labeling of mental illness. People who require psychological assistance have a low sense of self-efficacy after internalizing social representations of the public. To protect their more fragile self-esteem from further harm, they will use maladaptive coping strategies [26], such as avoiding interpersonal communication. Psychological counseling is an interpersonal communication process in which people seeking help must express their inner thoughts and underlying reasons in their subconscious to the counselor. Therefore, it is understandable that they would be concerned about a counselor looking down on them. As such, they avoid obtaining professional psychological help to reduce the social consequences.

### 4.2. High OIT Will Increase the WROC

Online psychological counseling requires clients to establish mutual trust with their counselors to reveal their feelings and personal privacy. Our results revealed a significant and positive correlation between the degree of OIT and WROC; that is, the higher the level of OIT, the greater the WROC, which supports Hypothesis 2.

This outcome can be understood from previous studies, which indicate a significant and positive correlation between OIT and online self-disclosure and that people with a high OIT are more likely to express their thoughts and feelings online [27,28]. In addition, online counseling is carried out through social media or social software, and people’s attitudes toward the use of social media or software may also affect their WROC. Liu and Liu [29] found that OIT positively predicted the acceptance and use of social media or social software; the higher the level of OIT, the more willing people were to use social media or software for interpersonal activities.

### 4.3. The Mediating Role of ECOC

Compliance with ethical norms is conducive to achieving counseling goals and aiding those who need help in actively seeking professional counseling services [30]. Our correlation analysis demonstrated that, on the whole, ECOC was significantly and negatively related to WROC. That is, the greater the ECOC, the lesser the WROC. When examining the specific dimensions, ethical concerns about online crisis intervention were significantly and positively related to more positive evaluations of online counseling. Hence, the greater the ethical concerns about access to crisis intervention, the more positive the evaluation of online counseling.

Although online psychological counseling does not require direct exposure to the counselor, clients still encounter ethical issues. The survey revealed that help-seekers’ concerns about the qualifications and identities of online counselors, the recording of the counseling process, the confidentiality of counseling records, informed consent, and timely help in the event of a crisis are serious. Online psychological counseling is conducted in a virtual space, and people who seek help may believe that its authenticity is lower than that of face-to-face counseling. In face-to-face counseling, clients can make a comprehensive assessment of the counselor and judge the counselor’s abilities and whether the counselor can offer a sense of security, but online counseling cannot lead to the same evaluation, which results in ECOC and reduced WROC.

Despite the limitations of online counseling, its openness, convenience, and affordability allow for access to crisis intervention services for those who require it, especially those who are unable to receive face-to-face counseling for various reasons. Online counseling provides them with a relatively safe space in which to obtain timely help and avoid unnecessary harm [31]. From this perspective, we can see that one’s concern about the ethics of online crisis intervention is associated with having positive attitudes regarding online counseling.

Further, our findings signal that the three dimensions of ECOC played a mediating role in the relationship between the self-stigma of help-seeking and the WROC; that is, the self-stigma of help-seeking both directly and indirectly affected WROC through the mediating effect of ECOC, thereby supporting Hypothesis 3. After internalizing the negative representations of patients with mental illness, help-seekers form a strong sense of self-stigma and become concerned that receiving psychological counseling will cause a disturbance in their normal lives. In addition, the transmission of information poses certain risks in the process of online counseling, and there is the possibility of privacy being violated at any time [32]; therefore, help-seekers have a high degree of concern about the ethics of online counseling, and the motivation to seek help from online counseling is weakened. Naturally, the WROC is also lower.

In addition, the three dimensions of ECOC played a mediating role in the relationship between OIT and online counseling discomfort; that is, OIT both directly and indirectly affected discomfort with online counseling through the mediating role of ECOC. Psychological counseling must facilitate a safe, stable relationship between clients and counselors, but individuals with a low OIT have greater uncertainty about forming ties with others online and a greater ECOC. Thus, they will be more uncomfortable with online counseling.

Our study indicates that we need to improve the ethical norms of online counseling and bring it under the supervision of the proper industry or departments, thereby ensuring compliance with the ethical norms of online counseling and enhancing individuals’ WROC. Simultaneously, taking full advantage of the convenience of online counseling and providing timely, effective psychological interventions will improve people’s opinion of online counseling, enabling them to further accept and recognize it.

### 4.4. Implications for Theory and Practice

From a theoretical perspective, the results, to some extent, indicate that college students’ willingness to receive online counseling is associated with self-stigma, online interpersonal trust, and online ethical concerns. Furthermore, the influence of these factors needs to be considered to enhance the value of online counseling applications and change the current situation of the insufficient willingness of college students to seek online counseling. Regarding practice, firstly, college students need to be educated and informed about the availability and expectations of online counseling. It would be important for them to understand that, in the internet era and in the context of the COVID-19 epidemic, online counseling is an important form of carrying out mental health services, and the unique advantages of online counseling can make up for the shortcomings of face-to-face counseling. Secondly, the low willingness of college students to receive online counseling could be addressed by counseling service providers countering the stigmatization of online counseling through explanation, publicity, and other ways of destigmatizing psychological help-seeking using online services. It would be especially important to ensure that the reception service staff, network technicians, and online counselors abide by the code of ethics for counseling and ensure, to the maximum extent possible, that clients’ personal information is not disclosed and is properly preserved. Where such services are offered, these ethical considerations need to be discussed with the potential client. In addition, training should be provided to online counseling personnel to ensure that counselors and receptionists are proficient in the operation of the online counseling platform. Finally, standardized operating procedures, supervision systems, and regulations for online counseling should be established to ensure that online counseling is operated and supervised accordingly and that the rights and interests of clients are not compromised owing to technical or other issues so as to maximize the protection of clients’ well-being.

### 4.5. Limitations and Future Research Directions

There are two major limitations to this study. First, the research design adopted was cross-sectional; therefore, causal relationships could not be determined. Second, we did not identify or measure participants’ mental health histories; thus, the results may not directly apply to people in need of psychological counseling. Future studies should analyze the willingness of real patients with mental illness in relation to online counseling through experiments or clinical observations. To address these limitations, future research can be conducted in the following ways: (1) longitudinal research tracking designs, such as conducting cross-lagged studies to extensively explain the relationship between self-stigma, online interpersonal trust, and ethical concerns about online counseling and their mechanisms of influence on the willingness to receive online counseling; and (2) conducting stratified studies by screening subjects with different degrees of psychological problems through relevant assessment tools, and verifying the generality and differences of the model in different groups of participants at different levels of functioning.

## 5. Conclusions

ECOC plays an important role in affecting the WROC. Formulating and improving online counseling ethical norms will not only deepen college students’ trust in online counselors but also effectively boost their WROC. In addition, guidance for reducing the self-stigma of help-seeking can both reduce ECOC and enhance WROC. Thus, in the internet era, under the premise that face-to-face counseling resources are insufficient and cannot meet students’ psychological needs, promoting the standardization and effective supervision of online counseling is vital to improving professional psychological services and preserving students’ mental health.

## Figures and Tables

**Figure 1 ijerph-19-16462-f001:**
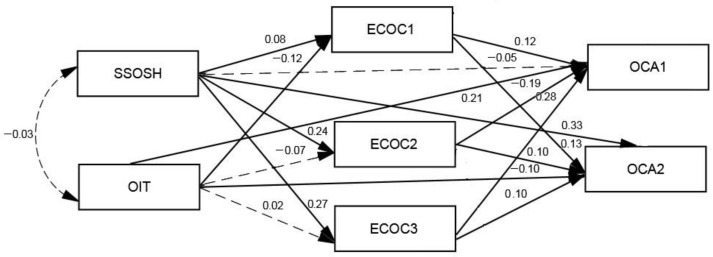
The standardized path map of the mediating effects of the dimensions of ethical concerns about online counseling. SSOSH, self-stigma of seeking help; OIT, online interpersonal trust; ECOC1, ethical concerns about online counseling (informed consent and confidentiality); ECOC2, ethical concerns about online counseling (the counseling relationship); ECOC3, ethical concerns about online counseling (crisis intervention); OCA1, online counseling attitudes (evaluation of online counseling); OCA2, online counseling attitudes (discomfort with online counseling).

**Table 1 ijerph-19-16462-t001:** Pearson correlations and descriptive statistics of the main study variables (*N* = 823).

Variable/Statistic	1	2	3	4	5	6	7	8	9
1. SSOSH	1								
2. ECOC1	0.08 *	1							
3. ECOC2	0.24 **	0.61 **	1						
4. ECOC3	0.27 **	0.28 **	0.43 **	1					
5. ECOC	0.20 **	0.90 **	0.81 **	0.62 **	1				
6. OIT	−0.03	−0.12 **	−0.07 *	0.01	−0.10 *	1			
7. OCA1	−0.02	0.05	−0.03	0.22 **	0.10 *	0.22 **	1		
8. OCA2	0.40 **	0.26 **	0.31 **	0.27 **	0.34 **	−0.13 **	−0.05	1	
9. OCA	−0.30 **	−0.15 **	−0.24 **	−0.05	−0.18 **	0.24 **	0.70 **	−0.75 **	1
*M*	2.54	4.19	3.76	3.04	3.77	3.38	4.11	3.32	3.89
*SD*	0.55	1.07	1.07	0.96	0.84	0.67	0.93	1.01	0.70
Skewness	0.28	−0.56	−0.14	0.05	−0.45	−0.06	−0.61	−0.08	0.20
Kurtosis	1.40	0.32	−0.28	−0.23	0.50	0.20	0.40	−0.32	0.28

Note. SSOSH, self-stigma of seeking help; ECOC1, ethical concerns about online counseling (informed consent and confidentiality); ECOC2, ethical concerns about online counseling (the counseling relationship); ECOC3, ethical concerns about online counseling (crisis intervention); ECOC, ethical concerns about online counseling; OIT, online interpersonal trust; OCA1, online counseling attitudes (evaluation of online counseling); OCA2, online counseling attitudes (discomfort with online counseling); OCA, online counseling attitudes. * *p* < 0.05; ** *p* < 0.01.

**Table 2 ijerph-19-16462-t002:** Linear regression of online counseling attitudes (OCA) on self-stigma of seeking help (SSOSH) and online interpersonal trust (OIT).

Dependent Variable	Independent Variable	*R* ^2^	*F*	*B*	*SE*	β	*t*
OCA		0.14	66.21 ***				
	Intercept			4.01	0.16		25.15 ***
	SSOSH			−0.37	0.04	−0.29	−8.87 ***
	OIT			0.24	0.03	0.23	7.05 ***

Note. *** *p* < 0.001.

## Data Availability

The data that support the findings of this study are available by emailing the corresponding author.

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
