# Peer review of "Factors Affecting Willingness to Receive Online Counseling: The Mediating Role of Ethical Concerns"

_ijerph, 2022, doi:10.3390/ijerph192416462_

Round 1

Reviewer 1 Report

The paper is very interesting and is about the current problem. The paper is quite short but contains every necessary element. The quality of the considerations presented in the article is very high. The introduction shows the background of the presented research and shows the formulated hypotheses. In Materials and Methods section there are presented all necessary informations about participants, measurements, and analysis. The results are presented in a short but sufficient version. Discussion and Conlusions sections are also correct.

I have one comment regarding the Likert scale. In Measurements subsection the author writes (lines 158-160) that "The original scale uses a 5-point Likert scale. However, to avoid the central tendency, we used a 6-point Likert scale...". Therefore, to measure online interpersonal trust the author uses 6-point scale. In my opinion, the 6-point scale is not good option because there is no safe "neutral" option. Besides, I see no consequence because the author also uses the 5-point scale to measure the self-sigma of seeking help.

I also have a note regarding the use of Pearson's coefficient, means and standard deviations. There are no information about the distribution of the variables. Therefore, it is unknown if the use of these measures is correct.

Reviewer 2 Report

I find great interest in the topic and paper you have created. However, several points may be needed to be applied so the manuscript will be enhanced. I look forward to your revisions soon so the paper may be available.

1. The abstract lacks the highlight of the problem needed to be addressed before the objective of the study. I would suggest adding this since it would engage readers to read the whole paper and gain the importance of the study.

2. The introduction provided the background, but the need for study is missing. I do believe that the study is relevant and important, but the problem is not evident for the need of study. This would help authors justify the need for study.

3. The literature section lacks related and recent studies in accordance to mental health mobile application availability, intention for usage, behavioral aspects in utilizing the mental health application. I think it would be of interest for you to read the study entitled "Utilizing Structural Equation Modeling–Artificial Neural Network Hybrid Approach in Determining Factors Affecting Perceived Usability of Mobile Mental Health Application in the Philippines".

4. I do not understand why a 6-point Likert Scale survey was utilized instead of the traditional 5-point and 7-point Likert Scale. This should be justified.

5. Why did you utilize correlation analysis when you considered structural equation modeling? What is the purpose when this analysis provides the direct effect already?

6. Provide implications both theoretically and practically. This would highlight the great contributions and applications of your study.

7. Provide limitations and future research direction.

8. The discussion and application should give suggestive implications.

I hope my comments and suggestions may help provide enhancement of the manuscript. 

Round 2

Reviewer 2 Report

The authors were able to address all concerns. Thank you very much.